# Hearing Health Awareness and the Need for Educational Outreach Amongst Teachers in Malawi

Grant Kapalamula [1], Kelly Gordie [2], Memory Khomera [1], J. Zachary Porterfield [3,4], Julia Toman [5] and Jenna Vallario [1,*]

1   African Bible College, Area 47, Lilongwe P.O. Box 1028, Malawi
2   Department of Audiology, School of Communication Sciences & Disorders, College of Behavioral & Community Sciences, University of South Florida, Tampa, FL 33612, USA
3   Department of Microbiology, Immunology and Molecular Genetics, University of Kentucky, Lexington, KY 40508, USA
4   Department of Otorhinolaryngology-Head and Neck Surgery, University of Kentucky, Lexington, KY 40508, USA
5   Department of Otolaryngology, Morsani College of Medicine, University of South Florida, Tampa, FL 33612, USA
*   Correspondence: vallarioj@abcmalawi.net

**Abstract:** Malawi, as a low-income country in southeastern Africa, severely lacks early identification, diagnosis and intervention measures for hearing loss. Due to its constrained resources, an educational awareness campaign targeted at professionals can be a cost-effective instrument in promoting good health care through awareness, prevention, and early identification of hearing loss. The aim of this study is to assess school teachers' knowledge of hearing health, audiology services, identification, and management of hearing issues before and after an educational intervention. Methods: A Pre-Survey, followed by an educational intervention, and a Post-Survey were completed by teacher participants. A similar World Health Organization-derived survey was also administered to compare to our locally adapted survey. Trends related to efficacy, performance, and survey improvement were evaluated. Results: A total of 387 teachers participated. The average score on the Post-Survey was significantly improved compared to the Pre-Survey (71% to 97% correct responses) with the educational intervention. The only predictive variable related to performance was the location of the school within the capital of Lilongwe compared to rural sites outside of the capital. Our locally adapted survey compared favorably to the WHO survey. Conclusions: The results suggest that there is a statistically significant improvement in the implementation of an educational program to increase the knowledge and awareness of hearing health care among teachers. Some topics were more poorly understood than others, suggesting the need for targeted awareness interventions. Location within the capital city had some effect on performance but a high rate of correct responses was achievable across the participants independent of age, teaching experience, or gender. Our data support the idea that hearing health awareness interventions can be an effective and low-cost option to equip teachers to effectively serve as an advocate for improved identification, early diagnosis and appropriate referral of students with hearing loss.

**Keywords:** global audiology; humanitarian audiology; hearing loss; pediatric audiology; Malawi; Africa; hearing health; hearing health awareness; hearing conservation; education





## 1. Introduction

Hearing loss is prevalent worldwide, with an estimated 34 million children suffering from deafness or hearing loss of which 60% of cases are due to preventable causes [1]. The burden of hearing loss falls disproportionately on lower- and middle-income countries [1]. The social and financial implications of hearing loss are profound, with an estimated global economic burden of USD 980 billion annually [1]. Malawi is among the poorest third-world

countries in the world. Located in southeastern Africa, Malawi has a growing population of 20 million people. Lilongwe, the nation's capital, is the most populous city; however, the urban population in Malawi is low, with 82% of the total population living in rural areas [2]. The birth rate is among the highest in the world, with an average of 4.3 births per woman [3]. Additionally, the nation struggles with significant poverty, with 51.5% of the population living under the poverty line and 20.1% living in extreme poverty [4].

According to the Global Burden of Diseases in 2019, the top six causes of death in Malawi are communicable diseases, and for the most part, these complications are preventable or treatable with modern medicine [5]. However, the quality of health care and the number of health care providers are scarce, with approximately two doctors and six nurses per 100,000 people [6]. For reference, in the United States, there are approximately 248 physicians per 100,000 people [7]. Access to hearing health care is even more challenging, with only two otolaryngologists and 13 audiologists in the entire country [8,9]. Many of the impacts of hearing loss can be reduced with early detection and intervention, but the lack of access to services results in an increased burden in these vulnerable populations [1].

Moreover, there are plenty of unknowns regarding the state of hearing health in Malawi, and generally, there is a lack of public awareness about hearing health. The prevalence of hearing loss nationwide has not been well elucidated and a centralized hearing screening program does not exist. A study conducted on the pediatric population of a rural community in Malawi found that the prevalence of bilateral hearing loss was as high as 12.5%, unilateral hearing loss 24.5%; and 46.9% of the sample had a potentially reversible etiology [10]. Furthermore, an estimated 29.5% of school-aged children with hearing loss are not enrolled in school [9]. However, given the lack of infrastructure, the scope of this assessment is narrow. Challenges to hearing health services are numerous and access to interventions in Malawi are limited [11,12]. As such, prevention and early detection are two of the most important tools available in the effort to combat hearing loss. Increasing public awareness and health education are cost-effective ways to efficiently and effectively utilize prevention and early detection of hearing loss and aid appropriate referrals [11,12].

The present study sought to assess primary school teachers' knowledge of hearing health, audiology services, and management of hearing issues before and after a hearing health focused educational intervention. This professional population was selected because of: 1. the burden of likely undiagnosed hearing loss in school-aged children in Malawi; 2. the significant challenges in identifying these individuals; and 3. the potential cost-effective benefit of increasing awareness among the individuals who are most engaged in the care of large numbers of children in Malawi. It is hypothesized that the teachers' knowledge will be limited but will improve following this educational intervention. With affirmation of this improvement, it will be evident that funding for such a hearing health training program for teachers is necessary to implement nationwide to improve hearing health care and education for all of Malawi's children and future citizens.

## 2. Materials and Methods

### 2.1. Overview

This study was undertaken in partnership with the African Bible College (ABC) Audiology team at the ABC Hearing Clinic and Training Centre (ABC HCTC). The educational intervention was developed by the ABC team in consultation with the University of South Florida (USF) Audiology and Otolaryngology collaborators. Data were collected in Malawi by the ABC team and sent for analysis to the USF team in the United States in close consolation with the team in Malawi. IRB approval was obtained from USF and Malawi's National Health Sciences Research Committee prior to undertaking this study. Permission for engagement of school teachers was also obtained from the District Education Managers for each school within the zones.

Teachers were surveyed in five school zones across Malawi: Dedza, Lilongwe Rural and Lilongwe Urban zones (Central West Division), and Salima and Ntchisi zones (Central

East Division). As the capital of Malawi, zones in the city of Lilongwe are considered urban relative to the rural zones of Dedza, Salima, and Ntchisi. A total of 25 schools were visited to give a diverse representation of the primary school teachers in the country. Gender, age, school name, school location, and teaching experience were recorded. The sample included approximately 20 primary school teachers selected randomly by the head teacher at the school. The sample of 20 teachers varied due to the school's size and teachers' individual schedules. Additionally, the sample was limited to primary school teachers within the five zones permitted by the District Education Managers and excluded student teachers, secondary school, tertiary school educators, and private institutions.

This study included a Pre-Survey questionnaire, an educational intervention, and a Post-Survey questionnaire. All surveys were administered on paper immediately before and after the educational intervention, and teachers had 30 min to complete each survey. Only data from teachers who completed both Pre- and Post-Surveys were included for analysis.

## 2.2. The Pre-Survey

The Pre-Survey was completed immediately before the educational intervention. The Pre-Survey included a Knowledge Survey assessment comprising ten true-or-false questions that measured the teachers' knowledge about hearing health, audiology, and hearing care as well as 14 additional true-or-false questions, taken from a World Health Organization (WHO) survey designed to assess hearing health knowledge (Appendix A, Table A1) [13]. The Pre-Survey also included four Likert-scale-type questions that were opinion based. These questions surveyed the teachers' points of view, such as how confident the teachers feel in their ability to identify a student with hearing difficulties or if they feel that further hearing care training is necessary.

## 2.3. Educational Intervention

The educational intervention was presented by two ABC HCTC Audiology students, Grant Kapalamula and Memory Khomera, who traveled to each school to provide optimum accessibility. The students were supervised by an audiologist (Au.D.) registered with the Ministry of Health in Malawi. The intervention was one hour long and completed in a lecture-style format via PowerPoint presentation (Supplementary Materials). Five main sections were covered in this educational intervention relevant to Malawi. Lessons covered in the lecture included the basics of ear anatomy, causes and degrees of hearing loss, the effects of hearing loss on speech and language acquisition and development, identification and signs of hearing loss in the classroom, hearing loss management and communication strategies for children, and the process of referring to nearby audiology services. All additional questions the teachers had for the ABC Audiology team were answered.

## 2.4. The Post-Survey

The Post-Survey repeated the Knowledge Survey and a new Opinion Survey comprising 4 Likert-scale-type questions assessing the teachers' response to the education outreach program.

## 2.5. Data Analysis

Results of the Opinion Surveys were analyzed by calculating the percentage of agreement and disagreement. The percent agreement included responses for "Strongly Agree" and "Agree" while percent disagreement included responses for "Strongly Disagree" and "Disagree".

Results of the Knowledge Surveys were analyzed by comparing percentages correct on the Pre-Survey and Post-Survey. The percent increase was calculated by dividing the difference in Pre- and Post-Survey scores by the Pre-Survey score. Performance on the surveys was compared based on variables thought to have a potential effect on the outcome, including age, sex, teaching experience, and location. Chi-squared analysis

was used for comparing the number of correct answers before and after the educational outreach. Comparisons between parametric groups were performed using a Student's T test while non-parametric comparisons were made using a Mann–Whitney test. Simple linear regression was used to compare performance on the WHO survey (Appendix A, Table A1) and our version adapted for the local context (Table 1). Multiple linear regression was used to evaluate dependence of survey score on independent variables of interest. Results were considered statistically significant if the resulting *p*-value was less than 0.05. Statistical analysis was performed using GraphPad Prism version 9.1.0 for Mac, GraphPad Software, San Diego, CA, USA, www.graphpad.com (accessed on 8 November 2022).

**Table 1.** Pre- vs. Post-Survey Knowledge Surveys by Question.

| Question | Pre-Survey Percent Correct | Post-Survey Percent Correct | Percent Increase |
|---|---|---|---|
| 1. Hearing loss can be of different severities and challenges. | 98% | 99% | 1% |
| 2. Ear infections can cause hearing loss. | 98% | 100% | 2% |
| 3. A hearing aid can be worn by a baby. | 24% | 86% | 259% |
| 4. Flus and sore throats can cause ear infections. | 56% | 95% | 70% |
| 5. Medication for TB, Malaria can cause hearing loss. | 59% | 95% | 62% |
| 6. Loud noises like Maize mills can cause hearing loss. | 82% | 99% | 21% |
| 7. Hearing loss can be inherited from parents. | 57% | 97% | 72% |
| 8. Diseases like meningitis, measles and mumps can cause hearing loss. | 75% | 99% | 33% |
| 9. Hearing loss can lead to inappropriate behavior in school. | 82% | 99% | 21% |
| 10. A child's poor academic performance may be caused by a hearing loss. | 78% | 99% | 27% |
| Total average | 71% * | 97% * | 37% * |

\* $p < 0.001$.

## 3. Results

### 3.1. Pre- and Post-Survey Comparison

A total of 387 teachers participated in this study, 69% of whom were female. The majority of respondents were between 30 and 50 years old. There was a diversity of teaching experience amongst the participants (Figure 1).

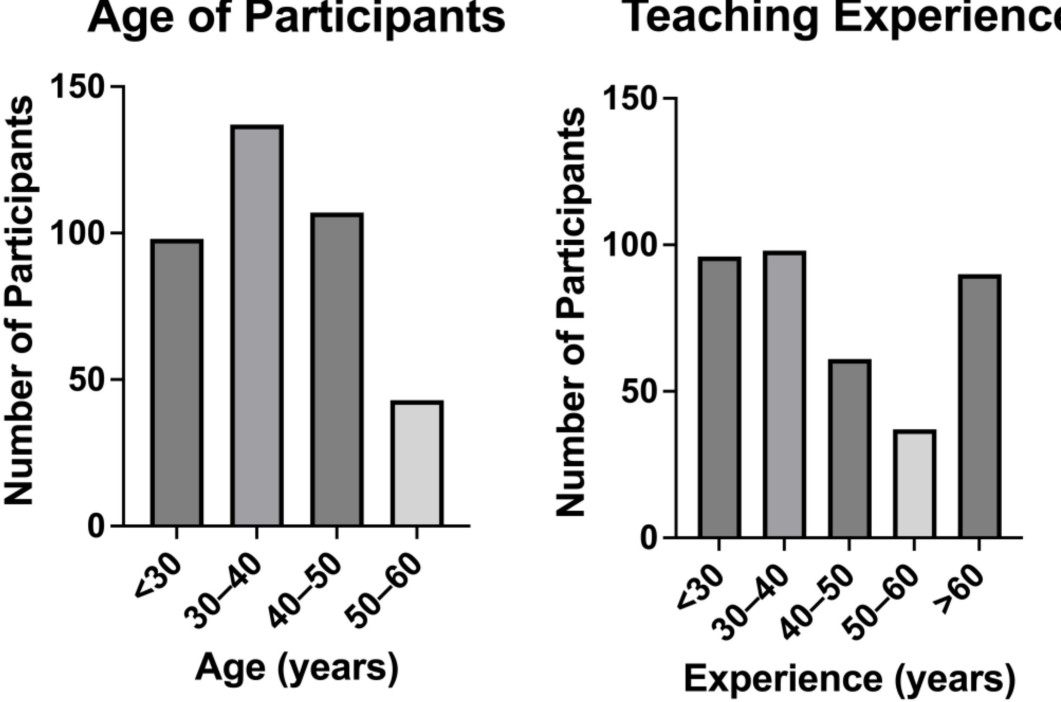

**Figure 1.** Age and teaching experience distributions of participants.

The average score on the Knowledge Pre-Survey was 71%, which increased to an average score of 97% on the Post-Survey (t = −30.951; *p* < 0.001) (Table 1); a 37% overall average increase. There was a range of knowledge highlighted by the Knowledge Survey.

Notably, some questions were missed more frequently than others, but all showed improvement after the educational intervention. Further, a majority of Post-Survey takers were able to answer the questions correctly. Question 3, evaluating knowledge about a baby's ability to wear a hearing aid, in particular, was missed more frequently than the rest, with a 24% pass rate on the Pre-Survey. However, this changed significantly following the intervention (Figure 2).

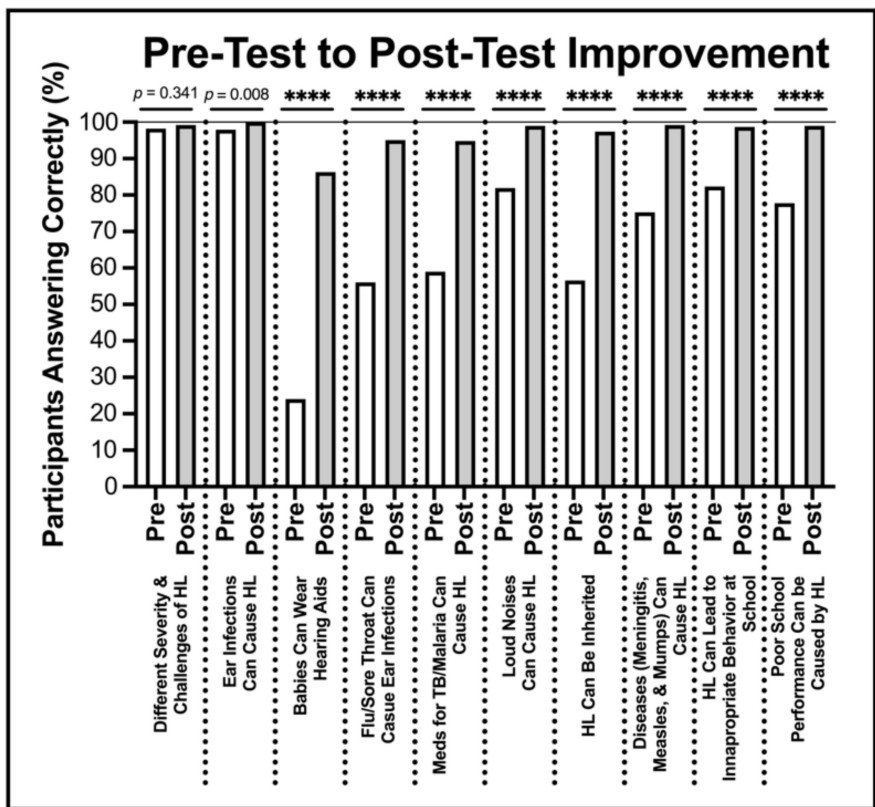

**Figure 2.** Pre-Survey to Post-Survey composite scores. There were ten survey questions before and after our hearing awareness intervention. The percentage of participants who answered these questions correctly are noted both before and after the intervention. *p* < 0.0001 indicated by ****.

*3.2. Lilongwe vs. Rural Sites*

The Knowledge Survey scores between schools in Lilongwe, the capital city of Malawi, and the other more rural sites located further from the capital were separately analyzed to compare the effect that site location had on the survey scores. There were 183 participants from schools in Lilongwe and 204 from sites not in Lilongwe. Pre-Survey scores for Lilongwe sites versus non-Lilongwe rural sites were 7.34 and 6.86, respectively (Figure 3).

The results of the Pre- and Post-Survey scores were compared by zone and then by school (Table 2). Kabango Primary in the Dedza district started with the lowest average score of all schools at 59%, but all teachers scored 100% after the intervention. The only other school that received 100% average on the Post-Survey Knowledge Survey was Mgoza in the Salima district.

**Table 2.** Pre- vs. Post-Survey Knowledge Surveys by District.

| District (n) | Pre-Avg Score | Post-Avg Score | Percent Increase | School (n) | Pre-Avg Score | Post-Avg Score | Percent Increase |
|---|---|---|---|---|---|---|---|
| Lilongwe Urban (102) | 76% | 96% | 27% | Mvama (25) | 78% | 98% | 26% |
| | | | | Msambeta (16) | 75% | 99% | 32% |
| | | | | Kabwabwa (21) | 70% | 97% | 37% |
| | | | | Lilongwe Demonstration (20) | 77% | 95% | 23% |
| | | | | Kalambo (20) | 78% | 93% | 20% |
| Lilongwe Rural (82) | 71% | 96% | 36% | Lilongwe Air Base (10) | 73% | 97% | 33% |
| | | | | Njewa Primary (15) | 75% | 94% | 26% |
| | | | | Likuni Girls Primary (18) | 73% | 98% | 35% |
| | | | | Chitedze (20) | 67% | 93% | 40% |
| | | | | Malingunde (19) | 65% | 96% | 47% |
| Ntchisi (61) | 69% | 97% | 40% | Phangwa (13) | 66% | 96% | 45% |
| | | | | Kalema (15) | 76% | 98% | 29% |
| | | | | Buzi (10) | 68% | 97% | 43% |
| | | | | Mpalo (15) | 68% | 94% | 38% |
| | | | | Madanjala (8) | 66% | 98% | 47% |
| Salima (76) | 68% | 98% | 45% | Mgoza (7) | 69% | 100% | 46% |
| | | | | Salima LEA (17) | 69% | 98% | 42% |
| | | | | MAFCO Primary (12) | 69% | 97% | 40% |
| | | | | Msalura Primary (25) | 69% | 98% | 43% |
| | | | | Kambwiri (15) | 64% | 99% | 55% |
| Dedza (66) | 68% | 98% | 44% | Kapesi (15) | 71% | 95% | 34% |
| | | | | Mwenji (12) | 70% | 98% | 39% |
| | | | | Kanama Primary (14) | 68% | 99% | 46% |
| | | | | Kabango (10) | 59% | 100% | 69% |
| | | | | Magaleta (16) | 73% | 99% | 35% |

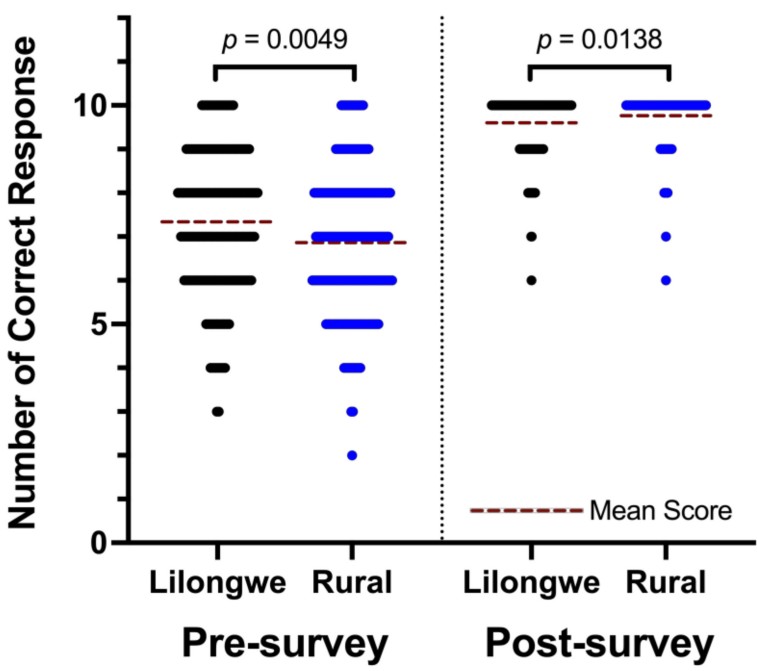

**Figure 3.** Lilongwe vs. rural sites (self-made questionnaire). Each dot represents the number of correct responses for an individual participant. Given the number of participants the dots overlap and the width of the resultant cluster is proportional to the number of respondents who answered that many questions correctly. The mean scores are represented by a red dotted line.

### 3.3. Opinion Surveys

The Pre-Survey Opinion Survey revealed that most teachers agreed that further hearing care training is necessary, with 97% agreement (Table 3). Most teachers disagreed (59%) that they have access to enough support for hearing issues at the nearby hospital (Table 3). Regarding whether the school can adequately help students with hearing difficulties, the teachers were essentially split between agreement (48%) and disagreement (47%) (Table 3). This result appeared largely dependent on the specific school, and upon further calculation of agreement per school, most teachers who agreed were from Lilongwe while the remaining teachers who disagreed were from more rural districts. In the Lilongwe Urban district, four out of the five schools surveyed reported a larger agreement that their school can adequately help students with hearing difficulties. However, in the rural district of Dedza, only one school out of the five surveyed had a larger average agreement: Kanama Primary (*n* = 14) at 50% agreement, 29% disagreement, and 21% undecided.

**Table 3.** Pre-Survey Opinion Survey.

| Question | Agreement (%) | Disagreement (%) | Undecided (%) |
|---|---|---|---|
| Further hearing care training is necessary. | 97% | 3% | 1% |
| My teacher training program prepared me for teaching a child with hearing loss. | 59% | 37% | 2% |
| The school can adequately help students with hearing difficulties. | 48% | 47% | 5% |
| We have access to enough support for hearing issues at the nearby hospital. | 29% | 59% | 12% |

The Post-Survey Opinion Survey revealed that 99% of teachers agreed that the educational intervention was helpful and that they need even more training (Table 4). It was agreed by 95% of teachers that they know what audiologists do, and 88% agreed that they could easily access services at the ABC HCTC (Table 4).

**Table 4.** Post-Survey Opinion Survey.

| Question | Agreement (%) | Disagreement (%) | Undecided (%) |
|---|---|---|---|
| This hearing training has been helpful. | 99% | 1% | 1% |
| We need more hearing training. | 99% | 1% | 2% |
| I know what audiologists do. | 95% | 5% | 0% |
| We can easily access services at ABC HCTC. | 88% | 7.5% | 4.5% |

On the Pre-Survey Opinion Survey, teachers were most confident in their abilities to identify students with hearing difficulties and their understanding of the impact of hearing loss on a child's education, at 88% and 87% agreements, respectively (Table 5). The teachers were less confident in their ability to manage a student with hearing difficulties at 53% agreement, even less teachers agreed that they knew how to prevent hearing loss at 44% agreement (Table 5). However, on the Post-Survey Opinion Survey, most teachers were confident in their knowledge and ability of all mentioned issues. The greatest improvement in percent agreement was observed for "I know how to manage a student with hearing difficulties" (a 65% increase to 88% agreement), and "I know how to prevent hearing loss" (a 121% increase to 97% agreement) (Table 5).

**Table 5.** Pre- vs. Post- Opinion Survey.

| Question | Pre- Agreement (%) | Post- Agreement (%) | Percent Increase |
|---|---|---|---|
| I can identify a student with hearing difficulties. | 88% | 97% | 10% |
| I understand the impact of hearing loss on a child's education and development. | 87% | 97% | 12% |
| I know how to manage a student with hearing difficulties. | 53% | 88% | 65% |
| I know how to prevent hearing loss. | 44% | 97% | 121% |

### 3.4. WHO vs. Malawian Self-Made Survey Scores

When comparing the scores of the WHO and Malawi Pre- and Post-Survey scores, the Pre-Surveys for the WHO and Malawi both have a distribution that centers around a mean of 9.21 out of 14 questions (65.8%) compared with 7.08 out of 10 questions (70.8%) while the Post-Survey average on the Malawi test was 9.69 out of 10 questions (96.9%). The difference between the Pre- and Post-Malawi Surveys is significant ($p < 0.0001$). The WHO test results were also normalized to the Malawi test results (divided by 14 and multiplied by 10) to make the scores directly comparable (Figure 4). The scores on the Malawi questionnaire were slightly better than the WHO questionnaire in a statistically significant way.

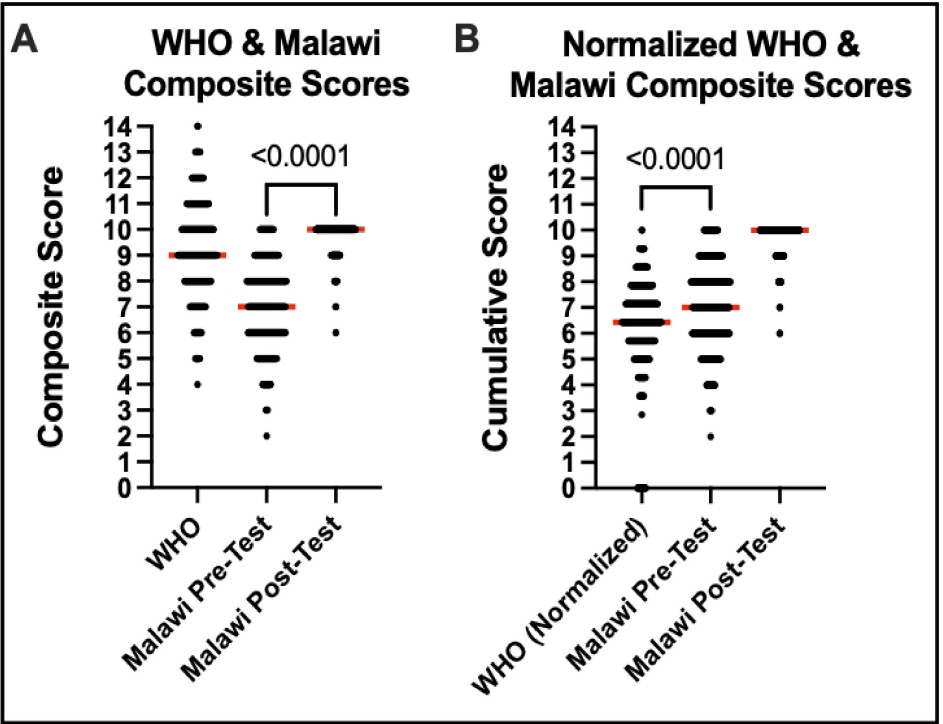

**Figure 4.** WHO-derived survey vs. Malawi-specific survey comparison.

Moreover, when comparing the WHO Pre-Survey composite scores to the Malawian Pre-Survey scores, the performance of participants between the standardized WHO Pre-Survey and the self-made Malawi Pre-Survey scores showed low-level correlation (Figure 5).

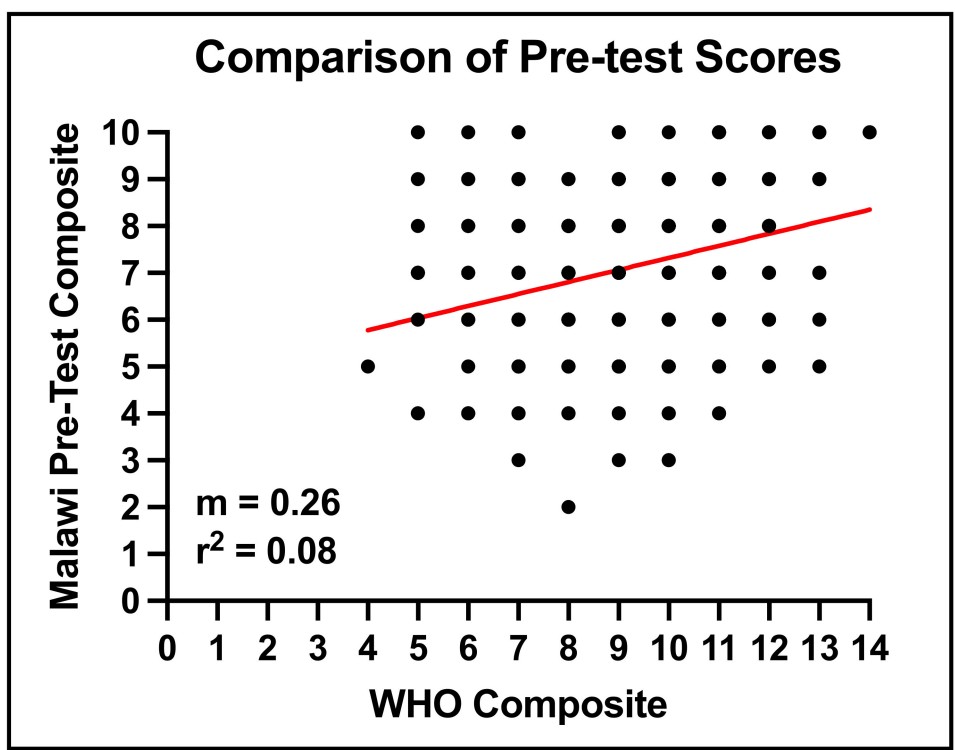

**Figure 5.** WHO-derived survey vs. Malawi-specific survey comparison. Each participant was graphed with the number of questions they answered correctly on the WHO questionnaire and the Malawi adapted questionnaire. The red line indicates the linear correlation between these data.

### 3.5. Multivariate Analysis

We also performed a multivariate analysis on the factors that could influence the odds of performance, including teaching experience, age, gender, and site (Lilongwe vs. other rural sites). Location in Lilongwe was the only variable of those tested that had a statistically significant effect on survey performance. Interestingly, participants located in rural sites were more likely to have better performance on the Malawi Post-Survey after the education intervention was completed (Figure 6).

### 3.6. Cross-Comparison of WHO Survey Scores

The WHO Survey (administered as part of the Pre-Survey, see Appendix A, Table A1) scores of our teachers were compared to two other studies which used the same WHO survey, from the general populations of Saudi Arabia and rural Indiana in the United States [14,15]. Compared to these other populations, our teachers scored lower overall (Table 6). This score was also lower than the self-made survey designed for this study.

**Table 6.** Comparison of the WHO Survey by Country.

| Country | Malawi Teachers (%) | Saudi Arabia [14] (%) | Rural Indiana, USA [15] (%) |
|---|---|---|---|
| Overall score | 61% | 76% | 67% |
| Infant hearing loss | 67% | 68% | DNA * |
| Cleaning and treatment | 58% | 80% | DNA * |
| Physical agents and over exposure | 55% | 68% | DNA * |
| Diagnostic delay | 86% | 86% | DNA * |

\* Data not available.

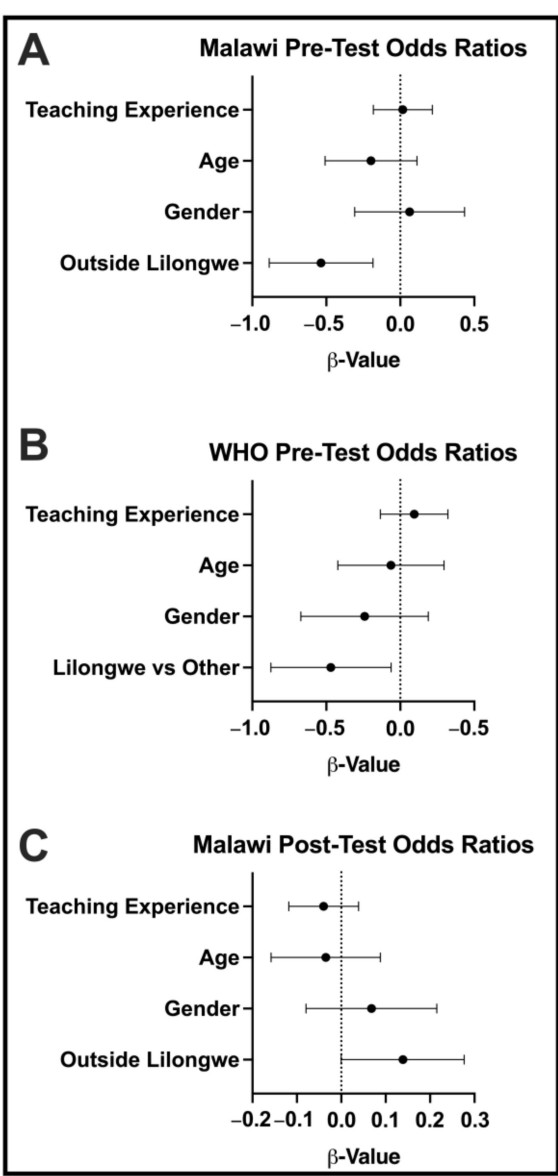

**Figure 6.** Multivariate analysis. The 95% confidence intervals are displayed.

### 4. Discussion

The specific aim of this study was to assess primary school teachers' knowledge of hearing health, audiology services, and management of hearing issues before and after an educational intervention. As hypothesized, teachers' level of understanding of some hearing topics was minimal but significantly improved following an educational intervention. This study's sample of 387 participants spread out across a varied setting is one the largest knowledge studies carried out in Malawi and similar sub-Saharan African settings. From an average Pre-Survey score of 71% to a 97% Post-Survey score (t = −30.951; $p < 0.001$), this study shows the value of educational awareness in improving hearing health literacy.

Expanding on the principles of prevention and awareness of hearing loss as cost effective strategies, Thompson et al. (2013) found that, in the United States, teachers are instrumental in the identification, prevention, and early detection of hearing loss in children [16]. It must be noted that the United States and Malawi have vastly different teaching environments and training qualifications [16]. In fact, Chataika et al. (2017) found that, in the Lilongwe Urban district of Malawi, most teachers claimed a severe lack of knowledge, skills, and resources regarding teaching children with disabilities [17].

However, the improvement of hearing health literacy by teachers has the potential for effective change in Malawi. A study by Mulwafu et al. (2017) in Malawi found that hearing health literacy among community health workers was 55%, and this was increased to 68% with a training intervention program [18]. Our intervention showed an even greater improvement in knowledge to 97%.

Hearing health training on teachers, nurses, or other professionals in contact with children can be a viable identification tool for referral to Audiology services in the country. This is especially important considering that conductive hearing losses, largely due to cerumen impaction and middle ear pathologies, are preventable and treatable conditions that individuals living in low-income countries often face [10]. Health education programs can assist in both raising the audiology awareness of the public and targeting professionals to aid appropriate referral and cost-effective intervention [11]. This, altogether, will contribute to the prevention of hearing loss in low-income countries such as Malawi. Effective synergy for hearing health education could be achieved with the utilization of new and existing training infrastructure such as the African Bible College Hearing Clinic and Training Center (ABC HCTC), and Save the Children Inclusion projects in schools, for example.

Moreover, this study's findings suggest that the impact of the educational intervention was most prominent in peripheral sites further from the capital city. Individuals from Lilongwe performed better on the Malawi Pre-Survey but participants from more rural sites located further from the capital city performed better on the Post-Survey in a small but statistically significant way. As the capital city, Lilongwe is the largest and best-equipped city in Malawi, with a greater presence of resource centers designed as a measure for helping children with different challenges in school. Therefore, teachers in Lilongwe may already have been exposed to some form of inclusivity training. Additionally, Lilongwe is the location of the ABC HCTC, and subsequently, teachers and schools within the clinic's catchment area may be more familiar with audiology and the signs of hearing loss in children, and ultimately feel more equipped to manage hearing loss. However, hearing health literacy can be significantly improved in these peripheral areas with limited access to resources as indicated by the current study. Indeed, 99% of teachers believe that further hearing care training is necessary, even after the intervention. The results of the current study provide further evidence that a hearing health training program has the potential for cost-effective disease identification and appropriate referral.

The most improved question from the Knowledge Survey regarded early identification of hearing loss: 3. A hearing aid can be worn by a baby. This statistic is a vital indicator of the severity in lack of knowledge of early intervention and viable remedial hearing devices such as hearing aids and cochlear implants especially in reference to infants and children. Although hearing aids, cochlear implants, and other remedial measures may currently be cost prohibitive in lower-income countries such as Malawi, the lack of diagnosis compounds the lack of demand for infrastructure development [12]. It also highlights the despondent state of infant hearing care and public awareness of congenital causes of hearing loss. The scarcity of resources that can be used for early identification and intervention are compounded by this lack of public awareness on hearing health care.

One aspect of this intervention which is not assessed by the current study is how durable the improvement in knowledge and awareness. Additionally, it is important to note that the surveys that were used have not undergone rigorous validation. The WHO survey is based upon materials developed by the WHO and focuses on key themes in prevention of hearing loss and overall hearing care. While this has been implemented in three separate publications, validation of this base survey and of the adapted Malawi version are important refinements to this work. Our group has planned follow-up investigations to reassess participants over time and to address survey validation more rigorously. There are some additional limitations to the study design, including some potential selection bias due to the randomization of participating teachers by school principals. Additionally, while the current study showed a significant improvement in hearing health awareness future studies could improve upon the intervention material to encompass other critical strategies

such as classroom noise management and the effectiveness of early identification and intervention in the classroom. Finally, longitudinal investigation is required to determine if the information gained remains as part of institutional memory or if ongoing educational efforts are required.

Overall, a comparison between the teachers in Malawi and the other general populations of Saudi Arabia and the rural areas of the United States of America shows a significant gap in hearing awareness. This is likely due to limited exposure to information, as the current study has proved an instantaneous increase in survey scores after exposure to the educational intervention. It is likely that the hearing awareness gap is even larger in the general public where average literacy rates are lower than those of professionals such as teachers. Moreover, cultural influences, which are more prominent in rural populations with lower literacy rates, have not been considered in this discussion.

## 5. Conclusions

Hearing loss is among the biggest antagonists to the success of a child in school and in future career development, even in well-developed countries. Improved identification and intervention for hearing loss can greatly assist in connecting hearing-impaired children to care, giving them a better chance of development and future success. Teachers stand at a unique intervention point for identification and referral of students and are a key target for these interventions. By providing teachers with the knowledge and skills to detect children with hearing loss, there now exists a viable means of giving these professionals the tools to aid identifying children in need and connecting them to appropriate care.

**Supplementary Materials:** The following supporting information can be downloaded at: https://www.mdpi.com/article/10.3390/audiolres13020024/s1, Educational Powerpoint.

**Author Contributions:** Conceptualization, G.K., M.K., J.T. and J.V.; methodology, G.K., M.K., J.T. and J.V.; formal analysis, K.G. and J.Z.P.; investigation, G.K., M.K., J.T., and J.V.; resources, G.K., M.K., J.T. and J.V.; data curation, K.G. and J.Z.P.; writing—original draft preparation, G.K., K.G. and M.K.; writing—review and editing, G.K., K.G., J.T., J.V. and J.Z.P.; visualization, K.G. and J.Z.P.; supervision, J.T. and J.V.; project administration, J.T. and J.V.; funding acquisition, J.V. All authors have read and agreed to the published version of the manuscript.

**Funding:** This research received no external funding. Private donations in support of the ABC Hearing Clinic and Training Center were used to support this work.

**Institutional Review Board Statement:** This study was conducted in accordance with the Declaration of Helsinki, and approved by the Ethics Committee of National Health Sciences Research Committee (protocol code #21/04/2701, date of approval 18 June 2021); USF Ethical review and approval were waived for this study due to non-human subject research for the data review portion of their contribution.

**Informed Consent Statement:** Informed consent was obtained from all subjects involved in this study.

**Data Availability Statement:** Not applicable.

**Acknowledgments:** We would like to acknowledge and thank the ABC Hearing Clinic and Training Center and the ABC college audiology department in Malawi. This project was made possible through the successful collaboration and partnership of two institutions: African Bible College and the University of South Florida.

**Conflicts of Interest:** The authors declare no conflict of interest.

## Appendix A

**Table A1.** WHO Questions.

| True or False Questions: Tick Any | True | False |
| --- | --- | --- |
| It is possible to diagnose deafness in infants shortly after birth | | |
| A deaf-mute cannot speak because of defects in the vocal tract | | |
| Hearing loss may cause attention deficits thus reducing school performance | | |
| Cotton buds are necessary for ear cleaning and are the safest means | | |
| Ear drops are sufficient to treat earache | | |
| Otomicosis (itchy ears) can be contracted at the swimming pool | | |
| Drug abuse does not provoke auditory hallucinations or modifications of hearing quality | | |
| Hearing aids need to fit accurately to provide maximum benefit | | |
| Kisses or slaps on the ears do not cause hearing problems | | |
| Listening to music for more than 3 h a day using earphones may cause permanent hearing loss | | |
| There are no tables recommending a reduction in the duration of exposure to high intensity noises | | |
| Irritating perception of sound (e.g., hearing metallic voices) and/or a reduction in hearing clarity (such as a sensation of having cotton wool in the ears) require medical advice | | |
| Sudden hearing loss is an emergency and requires an immediate audiological assessment | | |
| Age-related hearing loss may affect behavior | | |

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
