# Peer review of "Hearing Health Awareness and the Need for Educational Outreach Amongst Teachers in Malawi"

_audiolres, doi:10.3390/audiolres13020024_

Round 1

Reviewer 1 Report

Manuscript: A Survey of Teachers in Malawi, Africa: Hearing Health  Awareness

Audiology Research- 2212033

This study intended to investigate knowledge of hearing health, audiology services, and management of hearing issues in a group of primary school teachers in Malawi. Overall, it is a very useful work to explore an effective approach for improved identification, early diagnosis and appropriate referral of students with hearing impairment in low-income countries. In my opinion, the contribution is important in comparison to the previous studies. However, I would like to suggest the authors to consider the following issues. 

Major issues

  1. The article title needs to be revised. As far as I understand, the nature of this study is more important than raising hearing health awareness.
  2. Development of the survey questionnaires need to be described and justified appropriate. If it is a pilot study, it would be better to propose the future studies on how to validate the questionnaires properly.
  3. In my opinion, the Chi-square test seems more appropriate as a statistical analysis method to compare pre- and post differences for individual knowledge survey questions
  4. It is unclear about inclusion and exclusion criteria for the participants.
  5. The authors did not provide results relevant to the questionnaires for opinion survey presented in Tables 1 and 4.  
  6. In appendix, A-Tables 2 and 5 are the same, it would be more appropriate to keep one. Same comment to Tables 1 and 4
  7. Good to see that the authors provide the training materials in the supplementary document. Under the management section, I would like to suggest including the management of classroom noise, which is one of most relevant management strategies teachers who are able to implement directly. 

Minor issues

  1. There are two figure 2s.
  2. The first figure 2 is not needed, as far as I understand.
  3. The 2nd figure 2 is unclear. The lines and dots need to be described in detail.
  4. Graph 5 should be Figure 5. This figure needs further revision as well.
  5. Delete or revise the word of 'cost-effective' in the conclusion (line 326).

Overall, although this study has shown a certain degree of significance in terms of scientific value, unfortunately, the manuscript in its current state is not acceptable for publication and needs a major revision.

Author Response

We thank the reviewer for their detailed and insightful comments. We certainly feel that the revisions that have been made have strengthened both the manuscript and our message.

With regard to the comments from reviewer one, we summarize the changes we have made to our manuscript and our responses below:

See reviewer #1 comments below:

This study intended to investigate knowledge of hearing health, audiology services, and management of hearing issues in a group of primary school teachers in Malawi. Overall, it is a very useful work to explore an effective approach for improved identification, early diagnosis and appropriate referral of students with hearing impairment in low-income countries. In my opinion, the contribution is important in comparison to the previous studies. However, I would like to suggest the authors to consider the following issues. 

Major issues

  1. The article title needs to be revised. As far as I understand, the nature of this study is more important than raising hearing health awareness.

We agree with the reviewer that the significance of these data could be expressed more directly in the title. As such, we have updated it to “Hearing Health Awareness and the Role for Educational Outreach Amongst Teachers in Malawi”

  1. Development of the survey questionnaires need to be described and justified appropriate. If it is a pilot study, it would be better to propose the future studies on how to validate the questionnaires properly. 

Validation is a critical part of this process which was under-discussed in the manuscript. We thank the reviewer for pointing out this need. There are no currently validated surveys of this type and the WHO survey we employed was developed using WHO developed materials. While it has not undergone strict validation it has been used in two prior publications. The Malawi specific survey is based upon this material with slight adaptions to adjust it to the local context. As such, we agree that this is a pilot study and we are certainly in the process of setting up a follow-up

study with more formal validation process. We have added a section to this effect in the discussion. As follows:

“Additionally, it is important to note that the surveys that were used have not undergone rigorous validation. The WHO survey is based upon materials developed by the WHO and focuses on key themes in prevention of hearing loss and overall hearing care. While this has been implemented in 3 separate publications, validation of this base survey and of the adapted Malawi version are important refinements to this work. Our group has planned follow-up investigations to reassess participants over time and to address survey validation more rigorously.”

  1. In my opinion, the Chi-square test seems more appropriate as a statistical analysis method to compare pre- and post differences for individual knowledge survey questions 

This is an excellent point. We have added chi-2 analysis for each of the pre- and post-test questions and updated Figure 2 to show these values. All of the questions had a statistically significant improvement after the educational intervention, with the exception of the first question, which was correctly answered by the vast majority of the participants of a statistically significant difference would be hard to detect. We have also added some additional text to the Figure 2 legend to more completely describe this figure.

  1. It is unclear about inclusion and exclusion criteria for the participants. 

This has been updated to reflect the inclusion and exclusion criteria:

“A total of 25 schools were visited to give a diverse representation of the primary school teachers in the country. Gender, age, school name, school location, and teaching expe-rience were recorded. The sample included approximately 20 primary school teachers selected randomly by the head teacher at the school. The sample of 20 teachers varied due to the school’s size and teachers’ individual schedules. Additionally, the sample was limited to primary schoolteachers within the five zones permitted by the District Edu-cation Managers and excluded student teachers, secondary school, tertiary school edu-cators, and private institutions.”

  1. The authors did not provide results relevant to the questionnaires for opinion survey presented in Tables 1 and 4.  

We thank the author for pointing out an opportunity for clarification. Table 3 and Table 4 in the main text are the same as the Opinion Surveys listing in the Appendix. For clarity, we have removed the tables from the appendix as the questions they contain are listed in the main text tables and thus their duplication in the appendix does not add additional information. The discussion of these results is contained in the main text and is as follows:

“The Pre-Survey Opinion Survey revealed that most teachers agreed that further hearing care training is necessary with 97% agreement (Table 3). Most teachers disagreed (59%) that they have access to enough support for hearing issues at the nearby hospital (Table 3). Regarding whether the school can adequately help students with hearing difficulties, the teachers were essentially split between agreement (48%) and disagreement (47%) (Table 3).”

And

“The Post-Survey Opinion Survey revealed that 99% of teachers agreed that the educational intervention was helpful and that they need even more training (Table 4). It was agreed by 95% of teachers that they know what audiologists do, and 88% agreed that they could easily access services at the ABC HCTC (Table 4).”

  1. In appendix, A-Tables 2 and 5 are the same, it would be more appropriate to keep one. Same comment to Tables 1 and 4 

We thank the reviewer for pointing out the identical tables 2 and 5 and 1 and 4. As noted above, we have removed these for clarity as the questions are contained in the main text.

  1. Good to see that the authors provide the training materials in the supplementary document. Under the management section, I would like to suggest including the management of classroom noise, which is one of most relevant management strategies teachers who are able to implement directly. 

This is an excellent point. We have added the following paragraph to the discussion.

“Additionally, while the current study showed a significant improvement in hearing health awareness future studies could improve upon the intervention material to encompus other critical strategies like classroom noise management and the effectiveness of early identification and intervention in the classroom.”

Minor issues

  1. There are two figure 2s.
  • Thank you for noting this. We have adjusted the numbering

  1. The first figure 2 is not needed, as far as I understand.
  • We felt it was valuable to demonstrate that performance on individual questions varied widely but that post-intervention performance was universally quite good. While this point is made in the text, we feel the visual is helpful.

  1. The 2nd figure 2 is unclear. The lines and dots need to be described in detail.
  • We agree that this figure could be improved and have simplified it and also added additional detail to the figure legend.

  1. Graph 5 should be Figure 5. This figure needs further revision as well.
  • We have added additional text to the figure legend to improve interpretability of the figure.

  1. Delete or revise the word of 'cost-effective' in the conclusion (line 326).
  • This revision has been made.

Overall, although this study has shown a certain degree of significance in terms of scientific value, unfortunately, the manuscript in its current state is not acceptable for publication and needs a major revision.

Reviewer 2 Report

This is a well-written, informative, and sound manuscript.

I felt that the manuscript was very well written. 

Although concise, it addressed a critical issue regarding the lack of awareness, and thus, lack of appropriate treatment provided, for those experiencing hearing loss in Africa. 

The introduction and the conclusion underscored the severity of implications for that population.  Although the intervention was simple (teacher training via prepared lectures supplemented with PowerPoints) the pre-/post-test design clearly demonstrated causal effect.  Additionally, that same simple design lends itself well to replicability, as it can be easily accessed and carried out. 

This study can also be extended to other areas that may be experiencing a lack of hearing care awareness.

Author Response

Thank you to the reviewer for their time and consideration. No edits or review were required

Reviewer #2 comments:

This is a well-written, informative, and sound manuscript.

I felt that the manuscript was very well written. 

Although concise, it addressed a critical issue regarding the lack of awareness, and thus, lack of appropriate treatment provided, for those experiencing hearing loss in Africa. 

The introduction and the conclusion underscored the severity of implications for that population.  Although the intervention was simple (teacher training via prepared lectures supplemented with PowerPoints) the pre-/post-test design clearly demonstrated causal effect.  Additionally, that same simple design lends itself well to replicability, as it can be easily accessed and carried out. 

This study can also be extended to other areas that may be experiencing a lack of hearing care awareness.

Reviewer 3 Report

Dear Authors,

Thank you for the opportunity to review this interesting paper. The paper examines a survey of teachers in Malawi, Africa. The topic is interesting area. The manuscript is generally well-written. The study consists of a good dataset with 387 teachers.

Abstract:

Clear and well-written abstract.

Introduction:

Good introduction to the subject investigated. Very interesting and well-writen purpose in this study.

Materials and Methods:

Clear Overview and the review of the pre-survey, educational intervention and post-survey. Clear explanation of the data analysis method.

Results:

The study contains a lot of results and is clearly presented in text, tables, and figures. The study has an interesting comparison of the capital Lilongwe with the other sites. The intervention showed good results in all sites, especially in Kabango.

In 3.4. when comparing WHO and Malawian Self-Made Survey Scores, in Figure 3 – B needs some more explanation in text.

3.5. Multivariate analysis, add about this also in 2.5. in Data analysis. Interesting result about post-survey in areas outside Lilongwe (Figure 5).

Discussion:

Interesting discussion. The study highlights the most important findings in the discussion.

Conclusions:

An important study to achieve more awareness of the importance of audiology.

Author Response

We thank the reviewer for their review. No edits or revisions were required.

Reviewer #3 comments:

Dear Authors,

Thank you for the opportunity to review this interesting paper. The paper examines a survey of teachers in Malawi, Africa. The topic is interesting area. The manuscript is generally well-written. The study consists of a good dataset with 387 teachers.

Abstract:

Clear and well-written abstract.

Introduction:

Good introduction to the subject investigated. Very interesting and well-writen purpose in this study.

Materials and Methods: 

Clear Overview and the review of the pre-survey, educational intervention and post-survey. Clear explanation of the data analysis method.

Results: 

The study contains a lot of results and is clearly presented in text, tables, and figures. The study has an interesting comparison of the capital Lilongwe with the other sites. The intervention showed good results in all sites, especially in Kabango. 

In 3.4. when comparing WHO and Malawian Self-Made Survey Scores, in Figure 3 – B needs some more explanation in text. 

3.5. Multivariate analysis, add about this also in 2.5. in Data analysis. Interesting result about post-survey in areas outside Lilongwe (Figure 5).

Discussion:

Interesting discussion. The study highlights the most important findings in the discussion.

Conclusions:

An important study to achieve more awareness of the importance of audiology.

Round 2

Reviewer 1 Report

The authors have addressed all issues I raised. I am happy to suggest acceptance in present form for publication in Audiology Research.